# Generating ICS Anomaly Data Reflecting Cyber-Attack Based on Systematic Sampling and Linear Regression

**DOI:** 10.3390/s23249855

**Published:** 2023-12-15

**Authors:** Ju Hyeon Lee, Il Hwan Ji, Seung Ho Jeon, Jung Taek Seo

**Affiliations:** 1Department of Information Security, Gachon University, Seongnam-si 1342, Republic of Korea; 202240226@gachon.ac.kr (J.H.L.); ilhwan1013@gachon.ac.kr (I.H.J.); 2Department of Computer Engineering (Smart Security), Gachon University, Seongnam-si 1342, Republic of Korea; shjeon90@gachon.ac.kr; 3Department of Computer Engineering, Gachon University, Seongnam-si 1342, Republic of Korea

**Keywords:** industrial control systems, anomaly data generation, cyber-attack data, cybersecurity, machine learning

## Abstract

Cyber threats to industrial control systems (ICSs) have increased as information and communications technology (ICT) has been incorporated. In response to these cyber threats, we are implementing a range of security equipment and specialized training programs. Anomaly data stemming from cyber-attacks are crucial for effectively testing security equipment and conducting cyber training exercises. However, securing anomaly data in an ICS environment requires a lot of effort. For this reason, we propose a method for generating anomaly data that reflects cyber-attack characteristics. This method uses systematic sampling and linear regression models in an ICS environment to generate anomaly data reflecting cyber-attack characteristics based on benign data. The method uses statistical analysis to identify features indicative of cyber-attack characteristics and alters their values from benign data through systematic sampling. The transformed data are then used to train a linear regression model. The linear regression model can predict features because it has learned the linear relationships between data features. This experiment used ICS_PCAPS data generated based on Modbus, frequently used in ICS. In this experiment, more than 50,000 new anomaly data pieces were generated. As a result of using some of the new anomaly data generated as training data for the existing model, no significant performance degradation occurred. Additionally, comparing some of the new anomaly data with the original benign and attack data using kernel density estimation confirmed that the new anomaly data pattern was changing from benign data to attack data. In this way, anomaly data that partially reflect the pattern of the attack data were created. The proposed method generates anomaly data like cyber-attack data quickly and logically, free from the constraints of cost, time, and original cyber-attack data required in existing research.

## 1. Introduction

Industrial control systems (ICSs) are pivotal in critical infrastructures, like nuclear, thermal, and hydroelectric power plants, electrical grids, factories, and railways. Industry 4.0 has transformed ICSs by integrating advanced information and communications technology (ICT), enhancing efficiency and productivity but also elevating cyber threats [1].

Notable cyber-attacks, such as Stuxnet, Night Dragon, Duqu, Flame, Black Energy, NotPetya, Dragonfly, and Gas Pipeline Cyber-Intrusion, have targeted ICSs [2]. In response, security equipment deployment and cyber training have become essential for ICS cyber resilience [3]. However, obtaining anomaly data from cyber-attacks in real-world ICS environments, such as power plants, is challenging, necessitating research to acquire such data effectively.

The WADI (Water Distribution) testbed consists of a water distribution testbed and a water treatment system and is used for the security of the water distribution network [4]. WADI’s control network is connected to SCADA workstations through a wired or wireless network, and attack data are data generated after a cyber-attack is performed through the control network. An example of a cyber-attack is spoofing sensor readings to cut off the water supply to a consumer’s tank. Lei Chen and five others proposed a productive adversarial network (GAN)-based dual autoencoder for anomaly detection in the ICS called the DAGAN [5]. The DAGAN uses an encoder–decoder–encoder architecture to build a dual GAN model to learn the distribution of latent data. The DAGAN learns the generator and discriminator by generating benign and anomaly data in the learning process using a GAN. Phuc Cuong Ngo and five others proposed the Fence GAN, a better anomaly detection model, by modifying the loss function of a GAN [6]. KDD99 is a data set frequently used to study intrusion detection systems and consists of benign and cyber-attack network packets [7]. The KDD99 dataset, comprising benign and cyber-attack network packets, is crucial for developing intrusion detection systems, including a Fence GAN, which relies on attack data for learning. Building ICS testbeds through physical configuration, software simulation, and virtualization is vital for generating cyber-attack data [8]. However, these testbeds often struggle to mimic the diverse environments of power plants accurately. Therefore, AI-driven methods for generating cyber-attack data, independent of original attack data and using benign data, are increasingly necessary.

Generating data that accurately mirrors the characteristics of cyber-attacks involves several challenges. Firstly, it is crucial to ensure that the anomaly data possess distinct cyber-attack traits rather than being merely anomalous. This requires analyzing network packets from cyber-attacks to identify which features are unique to these attacks compared to benign data. For instance, packet count characteristics significantly indicate Denial of Service (DoS) attacks. Secondly, it is necessary to recalibrate various feature values to create new, credible data. Ignoring the intrinsic relationships between actual data features could lead to the generation of implausible data. Hence, a thorough understanding of the correlations among different data features is essential in generating realistic anomaly data.

This paper introduces a systematic approach to creating anomaly data that embodies cyber-attack attributes, utilizing benign data, systematic sampling, and linear regression models. The method comprises four key steps:Statistical analysis is first conducted to discern the differences between original benign and anomaly (attack) data. This step involves identifying distinct features through statistical analysis, which represents cyber-attack characteristics compared to benign data.The second step involves the creation of ‘contaminated’ data from benign data, embodying the previously analyzed cyber-attack characteristics. This is achieved by systematically adjusting the feature values of benign data, either increasing or decreasing them based on systematic sampling and probability distribution, effectively imposing cyber-attack characteristics onto benign data.The third, a linear regression model, is developed to generate new data, specifically anomaly data. The model learns to predict features by understanding linear relationships between data features. When ‘contaminated’ data are fed into the model, it is influenced by the altered feature values in the data, prompting it to modify the values and predict the remaining benign features. The data thus generated serve as potential anomaly data.The final step involves testing the newly created potential anomaly data using a classifier trained on the original benign and attack data. This helps determine whether the new data qualify as attack data. Among these candidate data, those identified as attacks are considered anomaly data that reflect cyber-attack characteristics.

The method of generating cyber-attack data presented in this paper offers several benefits over traditional approaches. Traditional methods often require significant time and financial investment in constructing a testbed. Even AI-based generation of cyber-attack data typically necessitates pre-existing cyber-attack data. In contrast, our method eliminates the need for such extensive resources and time commitment. It leverages benign data from the relevant domain, transforming it to create cyber-attack data, thus bypassing the limitations of conventional methodologies. This approach facilitates the easier creation of cyber-attack data by accurately incorporating attack characteristics. The contributions of this paper are as follows:In this paper, we performed a *t*-test to analyze the statistical difference between benign and attack data, from which we derived specific characteristics indicative of cyber-attacks. Subsequently, these identified characteristics were applied to the benign data through systematic sampling, effectively transforming them into cyber-attack (anomaly) data. These data were then generated using a linear regression model and validated with a classifier. The method introduced a four-step algorithm, integrating systematic sampling with a linear regression model, to generate and validate anomaly data that mirror cyber-attack characteristics;The proposed algorithm was applied to ICS_PCAPS data sourced from an ICS environment. The resulting anomaly data, reflecting cyber-attack characteristics, were successfully classified as an attack by the classifier trained initially on benign and cyber-attack data;Kernel density estimation was employed to analyze the generated anomaly data statistically. This analysis revealed a pattern transition in the anomaly data from the benign to the attack data category, indicating effective adaptation.

This paper is structured to provide a comprehensive overview and analysis. Section 2 delves into existing studies on generating anomaly data in ICS environments and discusses their limitations. Section 3 outlines the proposed algorithm for anomaly data generation. Experiments using ICS data to test and validate the algorithm are presented in Section 4. Section 5 addresses the limitations of the generated anomaly data, and Section 6 concludes the paper.

## 2. Related Work

In existing studies, anomaly data are generated using various methods and environments. In existing studies, anomaly data are generated for performance improvement and testing in IDS or anomaly detection systems. IDS and anomaly detection systems identify potential cyber threats by judging them as anomalies when they do not match benign patterns or detect anomaly patterns [9]. Data generation studies that can be used in an ICS environment include methods using AI and methods using testbeds. Anomaly data generation using AI mainly uses GAN to generate anomaly data. Table 1 overviews related work technology, data generation methods, data types, and whether cyber-attack characteristics are reflected.

### 2.1. Generating Anomaly Data Using AI

The G2D reduced the anomaly detection problem to binary classification to improve anomaly detection performance and make detection model implementation easier [10]. The G2D uses a GAN (Generative Adversarial Network), which trains two deep neural networks (generator and discriminator) using only benign data. Cases in which the generator failed to generate benign data during the learning stage were divided into the initial learning stage and the stage before complete optimization, and these stages were regarded as irregular generators. The G2D adopts irregularities at this stage as a method of generating anomaly (irregular) data. The G2D uses this method to generate anomaly data that deviates from the distribution of benign data. The TGAN-AD proposed a Transformer-based GAN for anomaly detection in time series data [11]. The generator of the TGAN-AD can improve performance because it can extract contextual features of time series data, and the discriminator is used to determine anomaly data. For the experiment of the TGAN-AD, an ICS data set such as SWaT (Secure Water Treatment) was used, and the GAN generated anomaly data and used it for anomaly detection. The MAD-GAN proposed GAN-based unsupervised multivariate anomaly detection using LSTM as the base model [12]. The MAD-GAN pointed out that unsupervised machine learning does not utilize correlations and other dependencies between various variables in the system when detecting anomalies. Instead of processing each data stream independently, the MAD-GAN considers the entire set of variables simultaneously and analyzes potential interactions between variables. This study created anomaly data using SWaT and WADI (Water Distribution) data containing attack data and perfectly utilized a GAN by using an anomaly score called the DR-score on the data.

### 2.2. Generating Anomaly Data Using Testbed

HAI data consists of benign and attack (anomaly) data generated from a Hardware-in-the-Loop (HIL)-based augmented ICS testbed [13]. The HAI testbed consists of a boiler, turbine, water treatment process, and Hardware-In-the-Loop (HIL) simulator to simulate various scenarios and environments. Attack (anomaly) data were generated after various cyber-attacks were performed on the HAI testbed. SWaT data are data generated from a water treatment test bed that produces 5 US gallons/hr of filtered water [14]. SWaT not only functions as a water treatment model but also provides a platform for studying potential cyber-attacks and their impacts and devising new countermeasures. Attack (anomaly) data are data generated by performing a cyber-attack on the SWaT testbed. An attack scenario using the SWaT testbed includes attacker A, who has access to the local factory communication network; attacker B, who is physically nearby but not directly at the site; and attacker C, who is at the site and has physical access to the device. SWaT attack scenarios consist of a system reconnaissance scenario, an infiltration scenario through a wireless network, and an infiltration scenario through physical access. This study instantiated the framework and implemented it with Python code. Attack data were generated after a cyber-attack on the DNP 3.0-based SCADA testbed was performed. Michael Dodson built an ICS honeypot and connected it to a global network to collect cyber-attacks that attackers accessed and carried out on the honeypot [16]. The ICS honeypot was built by imitating a Programmable Logic Controller (PLC) and a Remote Terminal Unit (RTU) and operated for 13 months. The ICS Honeypot analyzed the network collected over 13 months and found that only nine cyber-attacks were carried out. The cyber-attacks included data manipulation, buffer overflow, DoS, and retransmission attacks. Sinil Mubarak proposed a network traffic-based testbed using actual operating technology for CPS (Cyber Physical System) simulation and analysis [17]. In this study, a cyber-attack was performed using a cyber-attack tool provided by Kali Linux, and the resulting network traffic was collected. Afterwards, an intrusion detection system was developed using AI and the generated cyber-attack data. The Electra dataset was developed to evaluate the cybersecurity of electric traction substations used in the rail industry [18]. Electra data collect network packets resulting from cyber-attacks through attack implementation, traffic capture, features computation, processing, and labeling in a test bed of five PLCs.

Cyber-attack data are needed for ICS cybersecurity, but existing methods have several limitations and difficulties. Existing research uses GANs and testbeds to generate anomaly data. When generating anomaly data using a GAN, there is a limitation in that anomaly data are generated that are not related to cyber-attacks. Methods for generating cyber-attack data based on a GAN, such as the TGAN-AD and MAD-GAN, depend on cyber-attack data because they generate data like existing input data. In other words, methods using AI in existing research can only generate cyber-attack data if cyber-attack data are input. In order to generate anomaly data that reflect the characteristics of cyber-attacks, there is a method of performing and collecting attacks using a test bed. However, simulating a broad ICS environment with hundreds of assets or more is difficult and expensive. Additionally, the testbed does not reflect the actual environment settings. For example, if a power plant does not use the function code of a specific protocol, the data generated from the test bed are not practical. Additionally, even if you use honeypots to collect cyber-attack data, it is inefficient because you never know when an attacker will carry out an attack. Therefore, research is needed to create cyber-attack data using generated data to reflect the characteristics of the field domain.

## 3. Design of Anomaly Data Generation Structures

This section describes a method for generating anomaly data that reflects cyber-attack characteristics based on a linear regression model. First, Section 3.1 presents an overview of the anomaly data generation method, and Section 3.2 explains the *t*-test to analyze the difference between benign and attack data. Then, the features are selected through the *t*-test. In Section 3.3, we propose an algorithm that increases or decreases selected feature values through sampling techniques from benign data. In Section 3.4, we propose a linear regression model structure that generates anomaly data. Finally, in Section 3.5, we propose a verification method using a classifier to determine whether the generated anomaly data reflect actual cyber-attack characteristics.

### 3.1. Overview

Figure 1 schematically shows an overview of the anomaly data generation reflecting cyber-attack characteristics based on a linear regression model. The overview aims to analyze benign and attack data and generate anomaly data that reflect cyber-attack characteristics using only benign data. This method uses a *t*-test to analyze the difference between benign and attack data. It generates contaminated data (data with increased or decreased feature values reflecting cyber-attack characteristics) based on benign data through sampling. A linear regression model is a model that predicts each feature by learning the linear relationship between data features. If contaminated data are input into a linear regression model, other benign features are corrupted due to the contaminated features, creating anomaly data reflecting cyber-attack features. To this end, the generation of anomaly data reflecting the characteristics of cyber-attacks consists of the following four steps.

Data analysis and feature selection based on a *t*-test: statistically compare benign and attack data using a *t*-test. Through comparison, features that show a significant difference in t-score are selected. Afterward, only the selected features are chosen that are closely related to actual cyber-attacks. This feature reflects the characteristics of cyber-attacks;Generation of contamination data based on systematic sampling: Adjust the feature values of the benign data based on systematic sampling. The feature adjustment method increases or decreases based on the difference between benign and attack data in step 1. Feature increase or decrease is performed through systematic sampling and probability distribution;Learning linear regression model: Learn a linear regression model to generate anomaly data. Linear regression models learn linear relationships between features using only benign data. If the model uses contaminated data as the inputs, the remaining feature values are newly specified due to the features that reflect the cyber-attack characteristics in the contaminated data;Verification of cyber-attack data using classifier: Anomaly data candidates generated through a linear regression model are verified for attacks using a classifier. Only data classified as a cyber-attack through a classifier are anomaly data that reflect cyber-attack characteristics.

### 3.2. Data Analysis and Feature Selection Based on t-Test

A *t*-test is a statistical analysis method used to compare differences between two groups. Among the types of tests, Welch’s *t*-test was used. Other *t*-tests require that the two groups being compared have the same variance, but Welch’s *t*-test can be used on two groups with different variances [19]. Welch’s *t*-test must satisfy four conditions: identical interval and continuity, independence, normality, and unequal variance. A *t*-test has a null hypothesis that there is no difference between the two groups and an alternative hypothesis that there is a difference between the two groups. Welch’s *t*-test analyzes the differences between two groups by calculating the t-score and *p*-value. If the t-score is large and the *p*-value is less than the significance level (α) (usually 0.05), the alternative hypothesis is supported. The formula for calculating the *t*-score is as follows.
(1)$$\bar{X_{A}}=\frac{\sum_{i=1}^{n_{1}}X_{A,i}}{n_{1}}$$
(2)$$S_{A}=\sqrt{\frac{\sum_{i=1}^{n_{1}}{\left(X_{A,i}-\bar{X_{A}}\right)}^{2}}{n_{2}-1}}$$
(3)$$\bar{X_{B}}=\frac{\sum_{i=1}^{n_{1}}X_{B,i}}{n_{1}}$$
(4)$$S_{B}=\sqrt{\frac{\sum_{i=1}^{n_{1}}{\left(X_{B,i}-\bar{X_{B}}\right)}^{2}}{n_{2}-1}}$$

Equations (1) and (3) refer to the sample averages of the two groups (*A*, *B*). Equations (2) and (4) refer to the sample deviation of two groups (*A*, *B*). n is the size of the two populations.
(5)$$t=\frac{\bar{X_{A}}-\bar{X_{B}}}{S_{A-B}},\;S_{A-B}=\sqrt{\frac{S_{A}^{2}}{n_{1}}+\frac{S_{B}^{2}}{n_{2}}}$$

The differences between groups *A* and *B* can be compared by calculating the *t*-score. The *t*-score is the difference between the averages of two groups divided by the deviation of the two groups, as shown in Equation (5). The larger the absolute value of the *t*-score, the larger the mean difference between the two groups and the smaller the deviation. The formula for calculating the *p*-value is as follows.
(6)$$v=\frac{{\left(\frac{S_{A}}{n_{1}}+\frac{S_{B}}{n_{2}}\right)}^{2}}{\frac{{\left(\frac{S_{A}}{n_{1}}\right)}^{2}}{n_{1}-1}+\frac{{\left(\frac{S_{B}}{n_{2}}\right)}^{2}}{n_{2}-1}}$$
(7)p=T.CDF(t, v)

Since the statistical difference between the two groups cannot be determined with the *t*-score alone, the *p*-value must be calculated by calculating the degree of freedom (*v*). Equation (7) uses *T.CDF* to calculate the *p*-value. *T.CDF* represents the cumulative distribution function of the t distribution and gives the probability that a value of the *t* distribution with *ν* degrees of freedom is less than or equal to *t*.

The algorithm that uses the *t*-test to compare benign and attack data and select features that reflect cyber-attack characteristics is as follows.

Algorithm 1 identifies features that show statistically significant differences through the *t*-test between benign and attack data. First, the input uses benign data XNii=1L and attack data XAii=1L of data set X of size L. Afterward, XNii=1L and XAii=1L perform normalization and calculate the t-score and *p*-value for each feature through Welch’s *t*-test. If the t-score of the jth feature Fj is greater than the average of t-scores and the *p*-value is less than 0.05, the alternative hypothesis of the value is established. This means that Fj shows a significant difference between benign and attack data. This significant difference is between benign and attack data and is interpreted as a feature in which Fj reflects the characteristics of cyber-attacks. Finally, we analyze whether Fj is related to cyber-attacks and, if so, select features that reflect cyber-attack characteristics. For example, features related to DoS attacks and the total number of packets are highly correlated.
**Algorithm 1:** *t*-test and Feature Selection Algorithms.**Input:**  Benign Data: XNii=1L (Benign data set with data length L)Attack Data: XAii=1L (Attack data set with data length L)
**Output:** Feature where t is greater than t_avg, and p is less than α**Start**  n ← number of columns in XNii=1L (or XAii=1L)  n_list← an empty list  t ← an empty listp ← an empty list**For** j from 1 to n **do**XNfji=1L ← column j of XNii=1LXAfji=1L ← column j of XAii=1L   t, p ← Welch’s *t*-test (normalize(XNfjj=1n, XAfjj=1n))   n_list[j] ← t, p    t_sum += t**End**    t_avg = t_sum/n  **For** i from 1 to n **do**   α ←0.05  select feature ← an empty list   **if** n_list[i].t is greater than t_avg and n_list[i].p is less than α **then**    select feature [Fj] ← n_list[i]   **End**  **End**

### 3.3. Generation of Contamination Data Based on Systematic Sampling

Contamination data are generated using feature $F_{j}$ selected through a *t*-test. Contaminated data are generated by gradually increasing or decreasing the $F_{j}$ value of the benign data. If the t-score of the attack data $F_{j}$ is greater than the t-score of the benign data $F_{j}$, it is increased, and if not, it is decreased. Through this process, some characteristics of benign data become like those of a cyber-attack. The method for increasing or decreasing feature values was designed with inspiration from systematic sampling [20,21]. Systematic sampling basically extracts a specific sample using a certain pattern. The reason for generating contaminated data is to reflect cyber-attack characteristics in some characteristics of benign data and generate anomaly data that reflect cyber-attack characteristics. Contaminated data become the input to a linear regression model and are the first step in ensuring that the anomaly data generated by the linear regression model reflect the characteristics of cyber-attacks. The equation for performing systematic sampling is as follows.
(8)$$k=\frac{TN}{n},\;Sample=Random\;Strat+(i\;*\;k)$$

Systematic sampling calculates k by dividing the total number TN by the number of samples n and extracts samples with a constant value such as k. Systematic sampling has a random starting point. Contaminated data are attempted to increase or decrease using k, which is the average difference between attack and benign data divided by n based on systematic sampling. In this step, the random starting point was selected by applying a normal distribution to the characteristics of the benign data. As a result, multiple pieces of contaminated data were generated by gradually increasing or decreasing the $F_{j}$ value.

Algorithm 2 generates contaminated data using an algorithm using systematic sampling and a normal distribution based on benign data. Existing systematic sampling increases or decreases by a certain value based on a random starting point. In our proposed method, the starting point is the value obtained by applying a normal distribution to benign data, and the constant value is k. In our proposed sampling method, the starting points for each piece of contaminated data are random but increase or decrease by a multiple of k/γ. First, k is the average difference of each feature between the benign and attack data. Afterward, kR is calculated by dividing k by the number of contaminated data γ to be generated. Contaminated data are generated more than y increasing or decreasing by kR from the normal distribution applied to the benign data feature F. Contamination data generated through this process reflect cyber-attack characteristics and are used as inputs to a linear regression model.
**Algorithm 2:** Generating contaminated data Algorithms.**Input:**  Benign Data: XNii=1L(Benign data set with data length L)  Attack Data: XAii=1L(Attack data set with data length L)**Output:**  N Contaminated Data: X^Cii=1L**Start**    γ ← Number of contaminated data    Fj ← Selected features in step 1  **For** F from F1 to Fj **do**   normal_avg ← ∑i=1LXNFiL   anomaly_avg ← ∑i=1LXAFiL   k[F] ← anomaly_avg−normal_avg/γ  **End**  **For** i from 1 to γ **do**    XNewii=1L ← XNii=1L   **For** F from F1 to Fj **do**    μ ← calculating a mean XNewF      β ← calculating standard deviation XNewF    XNewFi=1L ← N(μ,β,len(XNewF))    kR ← k[F]      XNewFi=1L += i+1 ∗ kR      i th X^Cii=1L← XNewFi=1L  **End**  **End**

### 3.4. Learning Linear Regression Model

The linear regression model inputs contaminated data and outputs anomaly data candidates that reflect cyber-attack characteristics. In this step, the remaining feature values are affected and reconstructed by the features whose values have changed in the contaminated data (features that reflect cyber-attack characteristics). A linear regression model can predict features by learning linear relationships between data features [22]. Anomaly data candidates are created by applying the linear relationship between features of benign data to contaminated data ${\left[{\hat{X}}_{C}^{i}\right]}_{i=1}^{L}$.
(9)$$X_{N}^{f_{i}}={\sum_{j\neq i}^{p}\beta }_{j}^{i}X_{N}^{f_{j}}+\beta_{0}^{i}$$

Equation (9) expresses the linear relationship between the ith feature information $X_{N}^{f_{i}}$ of the benign data and the remaining feature information $X_{N}^{f_{j(-i)}}$. $\beta_{j}^{i}$ and $\beta_{0}^{i}$ are parameters of the linear regression model.

In Algorithm 3, a linear regression model learns linear relationships between features and generates anomaly data candidates using contaminated data as inputs. Algorithm 3 learns a forming regression model Mi that learns the linear relationship between p features and the remaining features, excluding the corresponding feature. Ω is the set of Mi. Generate anomaly data candidates X^Cii=1L using the set Ω of the learned model and the contaminated data X~Cii=1L. The ideal data candidate is the D-dimensional zero vector X~Cf, which is the predicted value of the linear regression model Mi with the contaminated data feature X^Cfi as input. This process is repeated as often as the number of contaminated data and the anomaly data candidate X~Cm is the output. Because X~Cm is a cyber-attack feature in the contaminated data, the remaining benign features are reconstructed into cyber-attack features.
**Algorithm 3:** Linear regression model training and output Algorithm.**Input:**  Benign Data: XNii=1L(Benign data set with data length L)  Contaminated Data: X^Cii=1L (Contaminated data set with data length L)**Output:**  Anomaly Data Candidate: X~Cii=1L**Start**  Ω ← empty set  p ← number of columns in XNii=1L (or XAii=1L)      **For** i from 1 to n **do**   Mi ← train the linear regression model with XNfj(−i)j=1L   Ω ← ΩUMi  **End**  **For** m from 1 to L **do**    X~Cf ← D demensional zero vector     X^Cfi ← contaminated data features   **For** i from 1 to p **do**       Mi ← get the i-th trained model from Ω    X~Cfi ← ∑j≠ipβjiX^Cfi+β0i (Mi-output)  **End**     X~Cf+=X~Cfi   **End**

### 3.5. Verification of Cyber-Attack Data Using Classifier

The classifier uses a classification model to verify whether the anomaly data candidate generated in the previous step is classified as a cyber-attack datum. The classification model classifies ${\left[{\tilde{X}}_{C}^{i}\right]}_{i=1}^{L}$ as benign or attack. The classification model is used because it is difficult to determine whether it is benign or an attack due to the uncertainty of the generated anomaly data candidate.

Algorithm 4 uses a classification model to collect data D that is determined to be actual attack data among anomaly data candidates X~Cii=1L. The algorithm determines p classification models and learns original benign and attack data. The learned classification model judges X~Cii=1L as an attack if it is in the same category as the actual attack data. The data determined to be an attack are anomaly data X~Aii=1L that reflect the characteristics of a cyber-attack.
**Algorithm 4:** Cyber-attack Classifier Algorithm.**Input:**  Anomaly Data Candidate: X~Cii=1L(Anomaly data candidate set length L)**Output:**  Anomaly Data reflecting cyber-attack characteristics: X~Aii=1L**Start**  ΩC ← empty set  p ← Number of classifiers  D ← empty set   **For** i from 1 to p **do**   Mi ← train the Classification Model with Original normal and attack data    ΩC ← ΩCUMi  **End**  **For** 1 to p **do**    **For** i from 1 to L **do**     yΩC ← ClassifierΩC(X~Cii=1L)      **if** yΩC is 1(attack) **then**     D ← DUX~Ci     **End**    **End**  **End**   X~Aii=1L ← D


## 4. Experiment and Evaluation

In Section 4, anomaly data reflecting cyber-attack characteristics were generated based on the algorithms and definitions described in Section 3. We evaluated the generated anomaly data and sought to answer the following research questions:(RQ1) Why can generated anomaly data reflect cyber-attack characteristics? (Section 4.3 and Section 4.4);(RQ2) Is there a change in the performance of the anomaly detection model if the generated anomaly data are used together with the original data? (Section 4.7);(RQ3) Can we determine that newly generated anomaly data correspond to actual cyber-attack data? (Section 4.6 and 4.7).

### 4.1. Experimental Setting

**Computing environment**: All experiments were performed in Intel(R) Xeon(R) Silver 4216 CPU 16 Core 2.10 GHZ, Tesla V100S-PCIE-32 GB 640(Tensor)/5120(CUDA) 1.23 GHz, 64 GB RAM, and 64-bit Ubuntu 20.04 LTS computing environments.

### 4.2. Dataset Description

The ICS_PCAPS [23] data set was used to generate anomaly data and train a classification model. ICS_PCAPS is widely used in experiments to detect cyber-attacks in ICS environments. ICS_PCAPS captured benign and cyber-attack packet data from a testbed consisting of a PLC, HMI, RTU, VFD (Vacuum Fluorescent Display), and 3-phase motor communicating based on MODBUS/TCP. ICS_PCAPS performed ICMP Flooding, TCP SYN Flooding, Modbus query Flooding, and a Main-in-the-Middle attack on the testbed and collected the generated network packets. In this paper, this experiment used 43,596 KB of benign packets and 35,104 KB of Modbus Query Flooding packets. Network packets were parsed, extracted features using CICFlowMeter [24], and converted to CSV files. Tise experiment used 11,146 benign data and 11,370 attack data converted to CSV.

### 4.3. t-Test Based Data Analysis and Feature Selection Experiment

In this experiment, benign and attack data were compared through Welch’s *t*-test, and features that reflect cyber-attack characteristics were selected. Before starting this experiment, missing values were removed from the benign and attack data, and features with only one duplicate value were removed. This was to improve the quality of the data set. Through this process, the data consisted of a total of 62 features. We normalized the data values to between 0 and 1 using Min–Max Scaling. We performed Welch’s *t*-test to compare normalized benign and attack data and to calculate the t-score and *p*-value. Among the t-scores and *p*-values for the two data, the feature in which the absolute value of the t-score is greater than the average is shown in Table 2.

The average of the absolute *t*-score value is 93.6087814. The features shown in Table 2 represent alternative hypotheses because the t-score value is much greater than the average, and the *p*-value value is 0. Therefore, the features in Table 2 indicate that they are statistically different. Figure 2 shows benign and attack data distribution for the top four features with large *t*-scores.

Figure 2 in this study presents a detailed data distribution analysis, with the x-axis representing data values and the y-axis indicating the density of data aggregation. The figure highlights that feature with high t-scores exhibit a pronounced difference in distribution between the benign and attack data. Specifically, features such as ‘tot_bwd_pkts’, ‘bwd_header_len’, and ‘subflow_bwd_pkts’, which share identical t-score values, demonstrate similar distributions in both benign and attack scenarios. In contrast, the feature ‘bwd_pkt_len_max’ stands out with a significantly higher t-score, indicating a more distinct distribution difference than the other features.

These variances in data distribution between benign and attack data indicate cyber-attack characteristics. Features that present values not typically observed in benign data are thus considered characteristic of cyber-attacks. For instance, features outlined in Table 2 are closely associated with the ‘modbusQueryFlooding’ attack. Some features, like ‘tot_fwd_pkts’, denote the total number of packets relevant in Flooding, a Denial of Service (DoS) attack characterized by generating many packets. A substantial increase in the total number of packets can indicate potential Flooding. Furthermore, features that track the number of bytes, such as ‘subflow_bwd_byts’, imply the transmission of a significant volume of data in the query response, another sign of Flooding.

Therefore, the features listed in Table 2 are pivotal for generating anomaly data, as they capture the essence of the specific attack patterns and behaviors observed in cyber-attacks. This analysis aids in creating more accurate and representative anomaly data, which is crucial for effective cyber threat detection and mitigation in ICS environments.

### 4.4. Systematic Sampling-Based Contaminated Data Generation Experiment

Contaminated data, in the context of this study, are generated by incrementally augmenting the values from a randomly selected starting point. This process is conducted through a normal distribution, employing a modified form of systematic sampling. The specific features of this procedure are sourced from benign data, as outlined in Table 2. First, to determine the amount of data increase, k is calculated by subtracting the feature average of the attack data and the feature average of the benign data. Afterward, calculate $k_{R}$ divided by γ, the number of contaminated data to be generated in k. Contaminated data are generated by gradually adding the features in Table 2 by $k_{R}$, based on the normal distribution applied to the benign data. In this experiment, γ was set to 10, and 12 contaminated data were generated, 2 more than γ. The reason why 12 pieces of contaminated data were created is because, assuming there are no actual attack data, the final value of the attack feature value is unknown. However, by analyzing cyber-attack characteristics based on existing research, it is possible to infer which characteristics reflect the characteristics of the cyber-attacks. Therefore, this method determines to what extent the values become like actual attack data when gradually increased. This method means that there is no limit to the number (increase) of contamination data to be generated. The data growth rate for the 12 contaminated data features is shown in Figure 3.

The amount of increased change through this experiment showed a more rapid rate of change as the gap between the benign and attack data increased, and the amount of data increase was derived randomly due to systematic sampling and normal distribution. As shown in Figure 3, if you improve the contamination data, features with similar gap values show a similar increase rate, but the more significant the gap value, the steeper the increase, so you can predict which feature will ultimately be the largest. Through this experiment, each contaminated data was generated based on 11,146 benign data, and some of the characteristics of the benign data reflected the characteristics of cyber-attacks. Since the contamination data were created based on benign data, there were 11,146 data per contamination data. This method can generate and utilize a large amount of data even if you have a small amount. The contamination data generated this way was used to generate the anomaly data as the inputs to a linear regression model. The answer to research question 1 is shown in Box 1.

Box 1Answers to Research Question 1(Answer to RQ1) We first identified statistical differences between benign and attack data. For statistical analysis, we used the *t*-test to analyze the difference between the two data statistically, and we identified the features that showed a statistically significant difference among the features of the two data. These differences are not visible in benign data but are caused by features that differ from the benign in the attack data. The identified features are shown in Table 2. These features refer to the total number of packets and bytes and are closely related to the Modbus Query Flooding attack. These features are cyber-attack characteristics that reflect the characteristics of cyber-attacks. Contaminated data are generated by adjusting the values of these features. Contaminated data are like cyber-attack feature values by increasing the values of some features of benign data through sampling techniques. These steps ensure that benign data reflect the characteristics of cyber-attacks. Contaminated data are used as inputs to the model for generating anomaly data, and the anomaly data generation model is influenced by the cyber-attack characteristics in the contaminated data so that other benign features are transformed to resemble cyber-attack data. In this way, new anomalous data reflect the characteristics of cyber-attacks.

### 4.5. Anomaly Data Generation Experiment Based on Linear Regression Model

A linear regression model learns linear relationships between feature information by learning benign data. The linear regression model was trained using 11,146 benign data. The linear regression model performs the prediction by looking at the remaining feature values for the feature to be predicted. Therefore, if contaminated data are input, the output data will be deformed due to the benign features affected by the cyber-attack features in the contaminated data. The data generated in this way are candidates for anomaly data. Anomaly data candidates can be generated as many as the number of contaminated data. In this step, the remaining features, except those in Table 2, are predicted. In other words, an anomaly (cyber-attack) data candidate is generated by predicting features other than the cyber-attack features in the contaminated data. This experiment generated 12 anomaly data candidates. The amount of data per anomaly data is the same as that of the contaminated data (11,145).

### 4.6. Cyber-Attack Data Verification Experiment Using Classifier

The classifier is used to verify whether the anomaly data candidate generated in the previous step is determined to be an attack. The models used in the classifier were Gradient Boosting, K-Nearest Neighbors, Logistic Regression, Random Forests, and One-Class SVM. Each classification model used 11,146 benign data and 11,370 attack data. Each classification model was optimized through hyperparameters. Each classification model generally judged more data to be attacks as the number of anomaly data candidates increased. Figure 4 shows the number of attack data identified by the classifier.

Anomaly data reflecting cyber-attack characteristics are classified as attack data by a classifier. Classification models such as Gradient Boosting and Random Forests classified more data as an attack as the number of anomaly data candidates increased. The K-Nearest Neighbors model classified the most attack data when the anomaly data candidate was seven, and after this, the number of attacks determined gradually decreased. Models such as Logistic Regression and One-Class SVM classified a similar amount of data as attack data for all anomaly data candidates. The five classifiers selected the candidate anomaly data most often classified as an attack, and only the data classified as an attack are the anomaly data that reflect cyber-attack characteristics. The generated anomaly data were created using the linear relationship of benign data through a linear regression model trained from the benign data, so they could be made using only benign data. At this stage, more than 50,000 anomaly data classified as attacks were generated.

### 4.7. Verification of Model Performance Using Anomaly Data

We want to verify whether the anomaly data generated through the proposed methods help the performance of an anomaly detection system targeting the ICS environment. In the ICS environment, it is difficult to improve the performance of the anomaly detection system due to a lack of anomaly data. We verify whether there is a change in the model’s performance by using the original cyber-attack data and the generated anomaly data together. The data used for training and testing is the ICS_PCAPS data set described in Section 4.2. The ICS_PCAPS data reflect the ICS environment because it is a widely used ICS protocol. The generated anomaly data set used anomaly data classified as attacks in anomaly data candidate 10. The training data used were 7801 original benign data, 7958 original attack data, and 2639 generated anomaly data. Gradient Boosting, K-Nearest Neighbors, Logistic Regression, and Random Forests were used as classification models, and hyperparameter tuning was performed. We used 3345 original benign data and 3412 original anomaly data for test data. The results before and after using the newly created anomaly data for learning are as follows.

The model’s performance was measured using indicators such as accuracy, precision, recall, and F1 score. These performance indicators determine how well the classification model classified the data [25]. K-cross validation was used to derive performance when only the original data were used. Because the ICS_PCAPS data used in this experiment were separated into different categories, the performance of the classification model was high even when only the original data were used. Therefore, when new anomaly data were used, the model’s performance was already at its maximum, making it impossible to improve performance further. However, the model trained with the new anomaly data does not show a significant performance degradation, so we can conclude that it is well classified as attack data and not in the ambiguous area between benign and attack data. The answer to research question 2 is shown in Box 2.

Box 2Answers to Research Question 2(Answer to RQ2) As a result of using the generated anomaly data in a classification model, there were no significant changes. There was no change because the data set used to generate the anomaly data was well-distinguished between benign and attack, so when only the original data were used, maximum performance was already achieved. Therefore, additional performance improvement was impossible when learning a model using new anomaly data because the performance was already at its maximum. However, the new anomaly data are classified as attacks in Table 3, and there was no significant performance decrease when the new anomaly data were used for learning, so the new anomaly data do not harm the model. If the new anomaly data were close to benign and trained with an attack label, the benign test data would have been classified as an attack, and the model’s performance would have deteriorated.

**Table 3 sensors-23-09855-t003:** Classification model results using newly generated anomaly data.

Performance Indicators (Before/After Using New Anomaly Data)		Gradient Boosting	K-Nearest Neighbors	Logistic Regression	Random Forests
Accuracy	Before	1.0000 (±0.0001)	0.9999 (±0.0001)	0.9999 (±0.0001)	1.0000 (±0.0000)
	After	1.0000	0.9999	0.9999	1.0000
Precision	Before	0.9999 (±0.0002)	0.9999 (± 0.0002)	0.9998 (± 0.0002)	1.0000 (±0.0000)
	After	1.0000	1.0000	0.9997	1.0000
Recall	Before	1.0000 (±0.0000)	0.9999 (± 0.0002)	1.0000 (± 0.0000)	1.0000 (±0.0000)
	After	1.0000	0.9997	1.0000	1.0000
F1 score	Before	1.0000 (±0.0001)	0.9999 (±0.0001)	0.9999 (±0.0001)	1.0000 (±0.0000)
	After	1.0000	0.9999	0.9999	1.0000

Kernel density estimation measured the distribution similarity between the original benign and attack data and the new anomaly data [26]. Kernel density estimation uses the kernel function to estimate the density of a variable. Non-parametric density estimation requires estimating density using purely observed data without prior information, and kernel function is one such method. Kernel density estimation is calculated using the following formula.
(10)$${\hat{f}}_{h}(x)=\frac{1}{n}\sum_{i=1}^{n}K_{h}(x-x_{i})=\frac{1}{nh}\sum_{i=1}^{n}K(\frac{x-x_{i}}{h})$$

In Equation (10), x is the input variable, and K is the kernel function. h is the bandwidth parameter of the kernel function and is a parameter that controls the smoothness of the graph shape. Kernel density estimation creates a kernel function based on the value of each observed data, adds up all the created kernel functions, and divides by the total number of data to create a graph. Below is a graph comparing the original benign and attack data with new anomaly data by applying kernel density estimation.

Figure 5 shows six representative graph types generated after applying kernel density estimation. In Figure 5, the x-axis represents the feature value, and the y-axis represents the density. Gaussian was used as the kernel function. The new anomaly data were created based on the original benign data, and because the linear regression model learned the benign data, the new anomaly data had the effect of adjusting from the original benign data to the original anomaly data in most graphs. This adjusted effect was found in totlen_fwd_pkts, idle_std, fwd_pkt_len_std, and bwd_iat_max in Figure 5. If there is no difference between the original benign and anomaly, such as pkt_len_mean, the new anomaly data are drawn in an almost similar pattern to the original data. Even if the original benign and anomaly data, such as flow_duration, were identical, in the case of features with a high correlation with cyber-attack characteristics, the value increased explosively due to the high correlation when generating data in the linear regression model, showing a completely different pattern. These graphs allow us to visually understand that although the new anomaly data are not entirely identical to the original anomaly data, the new anomaly data reflect the characteristics of the cyber-attack, resulting in a transformation from the original benign data to anomaly data. The answer to research question 3 is shown in Box 3.

Box 3Answers to Research Question 3(Answer to RQ3) The generated anomaly data have various characteristics and various values. As shown in Table 3, we used a machine learning-based classification model because the data interpretation was difficult due to data uncertainty. The classification model was trained on original benign and attack data. The classification model interprets various data features and analyzes whether the data falls into a category: benign or attack. In Table 4, most classification models classified anomaly data candidates as cyber-attack data as the number of candidates increased. This means that the generated anomaly data falls within the actual cyber-attack data category. The newly generated anomaly data were also statistically analyzed using Kernel density estimation. The new anomaly data were analyzed with the original benign and anomaly data, as shown in Figure 5. Since the new anomaly data are derived from benign data, in most graphs, the new anomaly data are adjusted from the original benign data to the original anomaly data. The characteristics of new anomaly data sometimes showed different aspects. However, the graph shows a pattern that resembles the original anomaly data. This situation means that the new anomaly data differ from the original anomaly data but reflect the characteristics of the attack data.

**Table 4 sensors-23-09855-t004:** Classification results by anomaly data candidates.

Anomaly Data Candidate Number and Result		Gradient Boosting	K-Nearest Neighbors	Logistic Regression	Random Forests	One-Class SVM
Anomaly data candidate 1	Benign	6293	11,128	5673	8423	5673
	Attack	4852	17	5472	2722	5472
Anomaly data candidate 2	Benign	5646	10,397	5649	7262	5649
	Attack	5499	748	5496	3883	5296
Anomaly data candidate 3	Benign	4969	7994	5684	6511	5684
	Attack	6176	3151	5461	4634	5461
Anomaly data candidate 4	Benign	3927	6386	5668	5617	5668
	Attack	7218	4759	5477	5528	5477
Anomaly data candidate 5	Benign	2537	5077	5619	4214	5619
	Attack	8608	6068	5526	6931	5526
Anomaly data candidate 6	Benign	1233	1102	5719	2922	5719
	Attack	9912	10,043	5426	8223	5426
Anomaly data candidate 7	Benign	484	281	5723	2029	5723
	Attack	10,661	10,864	5422	9116	5422
Anomaly data candidate 8	Benign	188	677	5828	1250	5828
	Attack	10,957	10,468	5317	9895	5317
Anomaly data candidate 9	Benign	96	1122	5801	568	5801
	Attack	11,049	10,023	5344	10,577	5344
Anomaly data candidate 10	Benign	89	2147	5737	196	5738
	Attack	11,056	8998	5408	10,949	5407
Anomaly data candidate 11	Benign	83	3196	5741	65	5741
	Attack	11,062	7949	5404	11,080	5404
Anomaly data candidate 12	Benign	97	3558	5866	29	5866
	Attack	11,048	7587	5279	11,116	5279

Our proposed method of generating anomaly data that reflect the characteristics of cyber-attacks can solve the limitations and difficulties that exist in existing research. Since there is no need to build a separate test bed first, there is no cost or time wasted, and data reflecting cyber-attack characteristics can be generated at any time as desired. In addition, existing AI-based anomaly data generation research requires original cyber-attack data to generate cyber-attack data. Still, the proposed method generates cyber-attack data based on benign data. Because the proposed method can utilize benign data generated from an actual domain, it is possible to generate anomaly data that reflect the characteristics of the actual domain, such as environment settings, assets, and networks. In other words, the proposed method generates cyber-attack data using real benign data without being hindered by constraints and difficulties such as cost, time, and dependency on cyber-attack data.

## 5. Limitations

The generation of anomaly data reflecting the cyber-attack characteristics proposed in this paper successfully created anomaly data to reflect the cyber-attack characteristics in benign network packets. However, the proposed method has some limitations.

Although the generated anomaly data reflect cyber-attack characteristics, they cannot perfectly imitate it. To generate data, we learned and created relationships between features based on benign data. However, some characteristics change the relationship slightly when a cyber-attack occurs. Although this research does not generate completely identical attack data, it is possible to generate data with a pattern adjusted from benign data to attack data, and it is possible to generate anomaly data with more diverse patterns.The proposed method requires an understanding of cyber-attacks. The network packets used in this experiment extract statistical features of the data through feature extraction. These features reveal the overall characteristics of network packets. It is necessary to understand cyber-attacks and predict features whose values will change rapidly when a cyber-attack is performed on the extracted and parsed features. Feature predictions can cite studies [27] analyzing existing cyber-attacks.

## 6. Conclusions

In Industrial Control System (ICS) environments, acquiring anomaly data stemming from cyber-attacks is crucial for multiple purposes, including testing security equipment and personnel training. However, it is difficult to collect such data in ICS environments. For this reason, this paper proposed generating anomaly data reflecting cyber-attack characteristics based on systematic sampling and linear regression models in the ICS environment. We outlined four distinct methods for this data generation process. The first method involves using the *t*-test to identify features significantly differing between benign and cyber-attack data. By examining the t-score and *p*-value, we determined which features most accurately represent the alternative hypothesis, thereby assessing their relevance to cyber-attacks. This step enhances the accuracy of identifying cyber-attack features in ICS environments. The second step involves creating contaminated data by modifying the benign data’s features. This modification is based on systematic sampling and the normal distribution patterns observed in the features previously identified. The third step applies the contaminated data to a linear model designed to learn and understand the linear relationships inherent in the features of benign data. In the fourth and final step, the anomaly data candidates generated with the linear model are fed into a classifier. This classifier is trained using the original benign and attack data. Only those candidates classified as attacks are selected for creating the final anomaly data set. This process ensures that the generated data accurately reflect the distinct characteristics of cyber-attacks, making it an invaluable resource for enhancing cybersecurity measures in ICS environments. In our experiment, we utilized the ICS_PCAPS data set based on the Modbus protocol. Welch’s *t*-test was employed for feature selection, leading to the generation of 12 sets of contaminated data through sampling and probability distribution. This approach yielded over 50,000 individual pieces of contamination data. These data were then used as inputs for a linear regression model, which generated candidates for anomaly data. Corresponding to the number of contaminated data sets 12 anomaly data candidates were produced, each comprising 11,145 data points. The subsequent stage involved using a classifier, which had been trained on the original benign and attack data, to process these anomaly data candidates. This classifier was responsible for categorizing the data into benign and cyber-attack classes. Anomaly data classified as indicative of a cyber-attack were deemed to reflect the characteristics of such an attack. Notably, each classifier yielded varying results for the anomaly data candidates. For instance, out of the 11,145 pieces of data in the anomaly data candidates, 105,407 were classified as cyber-attacks. To gauge the impact of this newly classified anomaly data on predictive models, the data deemed as attacks from anomaly data candidate 10 were repurposed as training data for four new classification models. This approach was aimed at understanding how the inclusion of new anomaly data influences the performance and accuracy of these models in identifying and responding to cyber-attacks within ICS environments. As a result, performance improvement was impossible because the performance was maximum when only the original data were used, but it did not lead to any significant performance decline. This means that the anomaly data generated based on benign data are closer to cyber-attack than benign and generate data that falls into one category. The distribution similarity of the original benign and attack data was compared with the new anomaly data using kernel density estimation. As a result of the distribution similarity comparison, the new anomaly data were created by modifying the benign data, so they did not perfectly imitate the original anomaly data. However, the new anomaly data pattern tended to change into the pattern of the original anomaly data. Although it is difficult to generate data that perfectly imitate the original attack data through the method presented in the paper, it is possible to generate data in the category of attack data. The proposed ideal data generation method can generate cyber-attack data without being hindered by the limitations and difficulties of existing work, such as cost, time, and dependency on cyber-attack data. In future research, we will improve the linear regression model structure through operations such as weight adjustment to reflect network packet characteristics better.

## Figures and Tables

**Figure 1 sensors-23-09855-f001:**
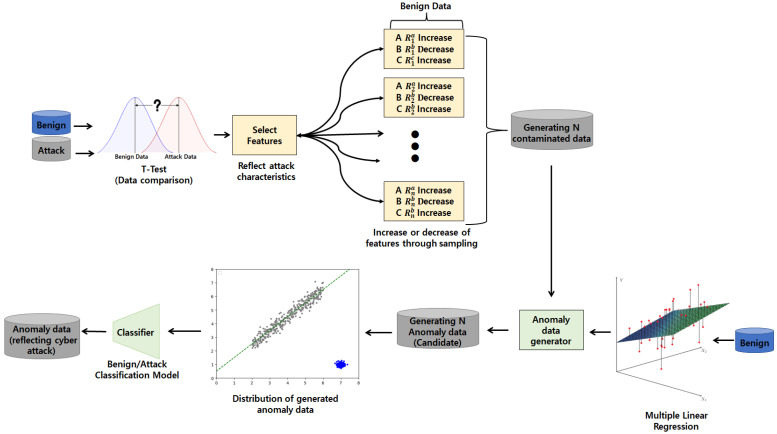
Overview of systematic sampling and linear regression model architecture for generating anomaly data reflecting cyber-attack characteristics.

**Figure 2 sensors-23-09855-f002:**
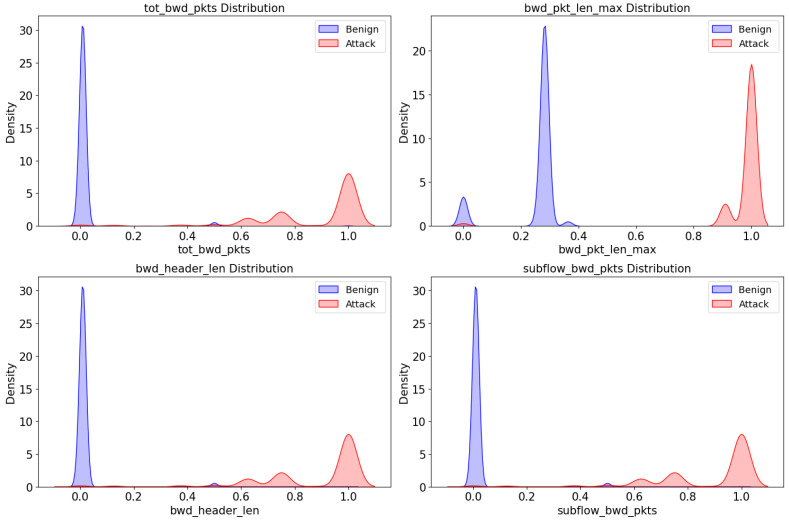
Comparison of distributions of features of the top four t-scores.

**Figure 3 sensors-23-09855-f003:**
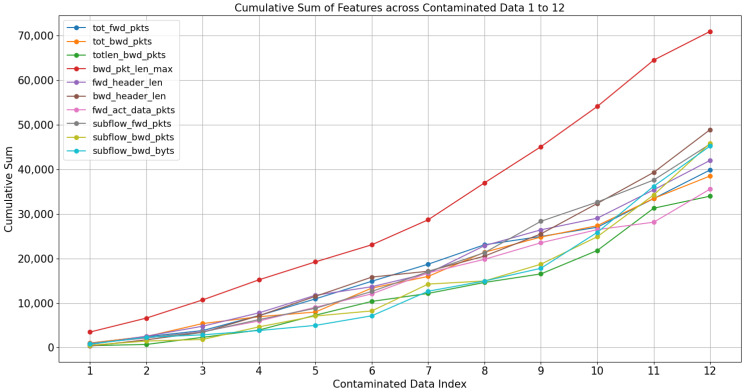
Increase in features for each contaminated data.

**Figure 4 sensors-23-09855-f004:**
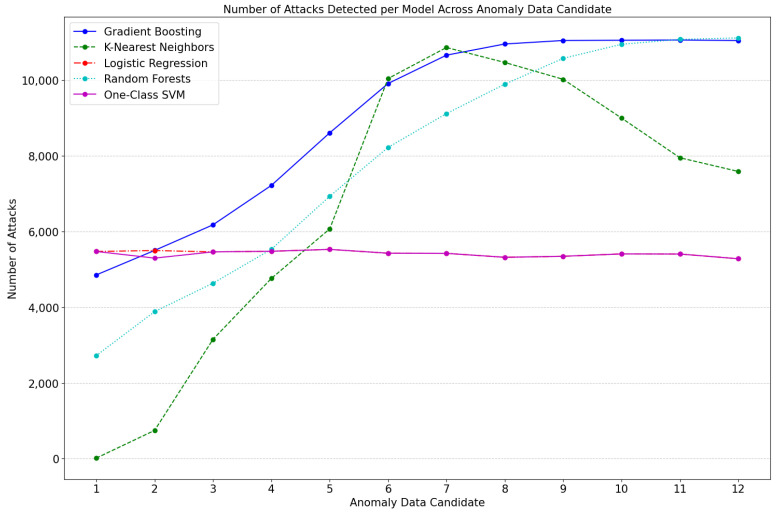
Number of attack data identified by classifier.

**Figure 5 sensors-23-09855-f005:**
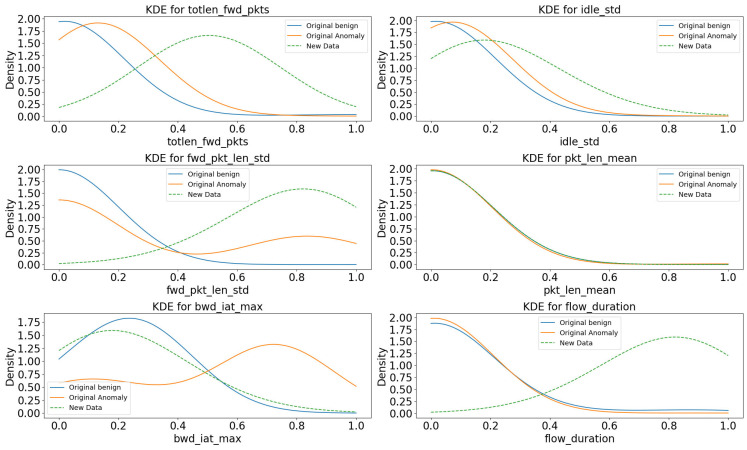
Six types of graphs created with Kernel density estimation.

**Table 1 sensors-23-09855-t001:** Overview of related work.

Ref.	Technique	Data Generating Methods	Data Types	Reflect Cyber-Attack Attributes
[10]	AI	Data generation using GAN	Image	No
[11]	AI	Data generation using GAN	Network Packet, Operational information	Depends on the dataset
[12]	AI	Data generation using GAN	Operational information	Depends on the dataset
[13]	Testbed (Hardware in the Loop	Cyber-attack using test bed	Operational information	Yes
[14]	Testbed (Water Treatment)	Cyber-attack using testbed	Operational information	Yes
[15]	Virtual testbed (DNP3 SCADA)	Cyber-attack using testbed	Network Packet	Yes
[16]	Testbed (ICS Honeypot)	Collection of cyber-attack data using honeypot	Network Packet	Yes
[17]	Testbed (Water Treatment)	Cyber-attack using testbed and attack tool	Network Packet	Yes
[18]	Testbed (Electric Traction Substation)	Cyber-attack using testbed	Network Packet	Yes

**Table 2 sensors-23-09855-t002:** Benign and attack data t-scores and *p*-values using Welch’s *t*-test. (Calculated as Benign-Attack).

Feature	*t*-Score	*p*-Value
bwd_pkt_len_max	−503.401	0
tot_bwd_pkts	−406.542	0
bwd_header_len	−406.542	0
subflow_bwd_pkts	−406.542	0
totlen_bwd_pkts	−390.317	0
subflow_bwd_byts	−390.317	0
tot_fwd_pkts	−213.986	0
subflow_fwd_pkts	−213.986	0
fwd_header_len	−211.599	0
fwd_act_data_pkts	−198.603	0
bwd_iat_tot	−174.77	0
bwd_pkt_len_std	−161.25	0
bwd_pkt_len_min	−144.118	0
down_up_ratio	−143.13	0
bwd_pkt_len_mean	−135.281	0
bwd_seg_size_avg	−135.281	0
pkt_len_std	−131.848	0
fwd_byts_b_avg	−127.83	0
bwd_iat_max	−101.139	0
idle_std	−94.2216	0

## Data Availability

Data are openly available in a public repository (2019). The data supporting this study’s findings are openly available in [ICS Cybersecurity PCAP repository] at [https://github.com/tjcruz-dei/ICS_PCAPS] (accessed on 10 December 2023).

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
