# Peer review of "Generating ICS Anomaly Data Reflecting Cyber-Attack Based on Systematic Sampling and Linear Regression"

_sensors, 2023, doi:10.3390/s23249855_

Round 1

Reviewer 1 Report

Comments and Suggestions for Authors While paper tackles topic with high practical value and is comprehensive enough, in general, there are many aspects which need significant improvement in order to proceed towards possible publication:  
  1. Introduction and abstract should put more emphasis on the advantages of the proposed approach compared to similar solutions
  2. Number of references has to be increased, considering both the similar methodologies and other relevant solutions which are comparable to the prospered one
  3. Related works section could also include additional tabular summary overview of similar approaches, together with underlying methods, features and covered scenarios
  4. Figure labels should be enlarged where possible as text is not clearly visible within them
Comments on the Quality of English Language

While there are no critical issues, the language style should be improved, in general. There are many repetitive phrases used in short distance between sentences. Additionally, there are some sentences which seem incomplete or out of context, so their connection with the rest of the text or previous content has to be improved, such as : "Future research will improve the linear regression  model structure to reflect network packet characteristics better. We want to create a model that can be used. " Please clarify such sentences and put them within the right context

Reviewer 2 Report

Comments and Suggestions for Authors

(1) The paper proposes the use of systematic sampling and linear regression methods to solve the problem of generating ICS abnormal data. The topic is innovative, the focus of the argument is prominent, and the experimental results are fully analyzed.

(2) There are duplicative sentences in the lines between 64-66.

(3) The data in the Figure 3 and Table 2 are inconsistent. In table 2 the data of Logistic regression and One Class SVM is almost the same, but in Figure 3, the anomaly data candidate 2 shows a different way.

(4) There are only two methods for generating anomaly data and their existing problems were provided in "Related Work" section. However, the manuscript doesn't offer the answer that whether systematic sampling and linear regression methods can solve these concerned problems.

Comments on the Quality of English Language

Normal

Reviewer 3 Report

Comments and Suggestions for Authors

1) The paper refers to normal data and attack data; referring to this data as benign and attack data could be clearer.

2) A paragraph to explain the purpose of the contamination data could be helpful.

3) The paper mentions Industry 4.0; a citation would be helpful.

4) The contribution statements in lines 100 - 119 are not clear and require improvement.

5) The Conclusion needs to be improved; lines 526 - 547 summarize the work performed but do not form part of the conclusion.

6) The paper answers research questions, but the original research questions are not provided.

Comments on the Quality of English Language

7) The English usage is colloquial in places and needs improvement. For example: "However, securing anomaly data in the ICS environment takes work" could be written as "However, securing anomaly data in the ICS environment is time-consuming".   "... and then inputs them into a linear regression model ..." could be written as " ... this transformed data is used to train a linear regression model ...". There are many instances where the English writing needs improvement.

8) Duplicated sentence, lines 65 - 66.

9) Lines 83 - 97, the four steps could be easier to read as bullet points.

Round 2

Reviewer 1 Report

Comments and Suggestions for Authors

Thank you for taking into account all the mentioned suggestions. The paper presentation quality now seems much better and can be accepted.

Reviewer 3 Report

Comments and Suggestions for Authors

I am happy with the authors response to my review comments.